# The Neuromodulatory Role of the Noradrenergic and Cholinergic Systems and Their Interplay in Cognitive Functions: A Focused Review

**DOI:** 10.3390/brainsci12070890

**Published:** 2022-07-07

**Authors:** Cody Slater, Yuxiang Liu, Evan Weiss, Kunpeng Yu, Qi Wang

**Affiliations:** 1Department of Biomedical Engineering, Columbia University, ET 351, 500 W. 120th Street, New York, NY 10027, USA; cs3791@columbia.edu (C.S.); yl3996@columbia.edu (Y.L.); ew2712@columbia.edu (E.W.); ky2449@columbia.edu (K.Y.); 2Vagelos College of Physicians and Surgeons, Columbia University, 630 West 168th Street, New York, NY 10032, USA

**Keywords:** locus coeruleus, the noradrenergic system, the cholinergic system, neuromodulation, norepinephrine, acetylcholine

## Abstract

The noradrenergic and cholinergic modulation of functionally distinct regions of the brain has become one of the primary organizational principles behind understanding the contribution of each system to the diversity of neural computation in the central nervous system. Decades of work has shown that a diverse family of receptors, stratified across different brain regions, and circuit-specific afferent and efferent projections play a critical role in helping such widespread neuromodulatory systems obtain substantial heterogeneity in neural information processing. This review briefly discusses the anatomical layout of both the noradrenergic and cholinergic systems, as well as the types and distributions of relevant receptors for each system. Previous work characterizing the direct and indirect interaction between these two systems is discussed, especially in the context of higher order cognitive functions such as attention, learning, and the decision-making process. Though a substantial amount of work has been done to characterize the role of each neuromodulator, a cohesive understanding of the region-specific cooperation of these two systems is not yet fully realized. For the field to progress, new experiments will need to be conducted that capitalize on the modular subdivisions of the brain and systematically explore the role of norepinephrine and acetylcholine in each of these subunits and across the full range of receptors expressed in different cell types in these regions.

## 1. Introduction

The central nervous system performs an incredibly large number of continuous computations, the result of which is to efficiently process the external world and execute a relevant response. In the human brain, an estimated 10^11^ neurons make approximately 1000 average connections to other neurons, forming up to 10^14^ distinct sites for information transmission. That is likely an order of magnitude higher than the total number of cells in the entire human body [1]. Even this staggering number of physical connections understates the complexity of information handling in the brain. Beyond simple neuron-to-neuron connections, multiple subtypes of glial cells are also known to play a role in synaptic transmission [2,3]. This ever-shifting structural background, across which the flow of information proceeds throughout an individual’s life, is then capable of giving rise to a diverse array of orchestral melodies through the 100+ endogenous substances that play a role in modulating synaptic transmission [4]. Some of these substances, known as neurotransmitters, can act over varying physical distances through their interaction with a much larger number of receptors.

Since the discovery of the first neurotransmitter, acetylcholine, in 1926 by Otto Loewi, there has been an explosion in the identification and understanding of chemical neurotransmission. Conceptually simplified, information transfer occurs in two modes: electrical propagation within neurons or chemical propagation outside neurons. Neurotransmitters are the chemicals that traverse the physical division between cells connecting the postsynaptic cell with information from the presynaptic cell. This is primarily mediated through an array of specific receptors on the postsynaptic cell. For any given neuron, the combination of presynaptic inputs will determine if a message is electrically transcribed and transmitted. If it is transmitted, an action potential will travel down the length of a neuron, resulting in the release of extracellular neurotransmitter onto the dendrites of postsynaptic cells. The substances released are usually tightly regulated and reuptaken or degraded to limit the action of the substance on its target.

Generally, neurotransmitters may have excitatory, inhibitory, and neuromodulatory effects on neurons through the action of their receptors. Excitatory receptors, when activated by corresponding neurotransmitters, result in a membrane depolarization and the propagation of an action potential. Glutamate is the neurotransmitter that predominately mediates excitatory effects through its receptor, which is nearly ubiquitously expressed in all types of neurons and many types of glial cells [5]. Inhibitory receptors exert an opposing effect, with the binding of corresponding neurotransmitters resulting in a membrane hyperpolarization that limits the ability of a neuron to initiate an action potential. In the mature brain, γ-aminobutyric acid (GABA) is the primary neurotransmitter that exerts inhibitory effects on neurons through GABAergic receptors [6]. The interplay between these two competing systems has been studied in a variety of contexts [7,8,9] and provides the foundation for how neurotransmission is thought to occur in the brain. The summation of excitatory and inhibitory inputs at every connection point in the brain determines the direction and pattern of information propagation in complex networks of neurons. The third type of neurotransmitters, known as neuromodulators, add an additional, but important, complexity to this paradigm by altering the balance of transmission on a micro, meso, or macroscale.

There are four primary neuromodulatory systems: acetylcholine, norepinephrine, dopamine, and serotonin. Each of these four molecules plays an important function in altering basic synaptic transmission patterns. The groups of neurons responsible for the delivery of these four neuromodulators are known as ascending neuromodulatory systems since each of these neurotransmitters originates in the brainstem, midbrain, or basal forebrain and projects to various brain structures. While early work characterized these ascending modulatory systems as highly collateralized and largely exerting their influence through the global regulation of neural activity [10,11], more recent work has emphasized the subdivision of these systems into cortical-region-specific sub-systems that can differentially influence information processing [12,13]. Subsequently, there has been a shift away from viewing neuromodulatory systems as mere “state-setting” or “gating” systems toward a framework in which there is also an understanding that a subset of highly specific neuromodulatory projections to the forebrain, especially the cortex, are necessary for specific cognitive functions [14].

This review focuses on the role of two of these neuromodulatory systems, the noradrenergic and cholinergic systems, in modulating cognitive functions. Acetylcholine (ACh) is an important neuromodulator that has long been implicated in learning, memory, and synaptic plasticity [15,16]. More recent work, however, has also highlighted the role of ACh in attentional effort, orienting, and the detection of behaviorally significant stimuli [17,18]. Norepinephrine (NE) has classically been viewed as a major mediator of arousal that plays an important role in regulating the efficiency of external sensory processing [19,20,21,22,23,24,25]. Recent work expanding on this has revealed a varied and complex role for the noradrenergic system in everything from memory formation [26] to executive function and attention [27,28], to cognitive flexibility [29,30], and to decision-making [31,32,33]. Dysfunction in either of these two neuromodulatory systems or in the coordinated interaction between them is heavily implicated in numerous neurodegenerative and neuropsychiatric diseases [34,35,36,37,38,39,40,41].

Understanding the complex and dynamic role of neuromodulation on sophisticated behavioral outcomes requires not only an understanding of the full range of impact of each neuromodulator and its various anatomically distinct subdivisions but also an understanding of the interplay between multiple neuromodulatory systems. These interactions could occur at the functional level, seen in an organism’s behavior; the anatomic level, through connectomic communication; the cellular level, through intertwined signaling pathways; or even at the synaptic level, when important brain regions experience the release of multiple neurotransmitters. A more complete understanding of the role neuromodulation plays in brain functions will require a better understanding of when and how two systems, such as the cholinergic and noradrenergic systems, collectively work together. The implications will be critical for better understanding and predicting normal brain function and for providing novel treatment strategies for aberrant brain function.

## 2. Anatomical Overview

### 2.1. Cholinergic System

#### 2.1.1. Sources

The cholinergic system is primarily comprised of groups of cells in the basal forebrain and midbrain that send diffuse but sparse projections to the rest of the brain [42,43,44,45,46,47,48,49,50,51,52]. Only the striatum differs with a local supply of cholinergic neurons for local transmission [53]. In primates, the cholinergic input to the cerebral cortex originates almost entirely from the nucleus basalis of Meynert (NBM), located in the basal forebrain [54]. These neurons are large with extensive dendritic trees, and a single neuron can innervate multiple brain regions, though there is minimal overlap in the axonal fields [50,55]. While cholinergic projections are widespread throughout the cortex, multiple studies have shown that there is a distinct pattern of segmented innervation depending on which nuclei in the basal forebrain the cholinergic neuron originates from [17,51,56,57,58].

A second major cholinergic source is found in the continuous array of multipolar neurons in the medial septum (MS) [46] and diagonal band of Broca (DB) [59,60]. A whole-brain atlas of projections from these regions in mice has found that cholinergic neurons were distributed in an uneven pattern between different brain regions, with neuronal density varying more than 10-fold across various structures [50]. Additionally, soma volume of cholinergic neurons varied by up to 8-fold across the examined regions [50]. Taken together, this work has demonstrated the existence of region-specific subtypes of cholinergic neurons that perform varying functions, though unlike in the nucleus basalis, there does not seem to be a specific relationship between soma location in the nuclei and the location of projections [50].

#### 2.1.2. Inputs

Cholinergic neurons in the nucleus basalis (NB) receive dopaminergic input from the ventral tegmental area and substantial nigra, serotonergic input from the raphe nuclei, and noradrenergic input from the locus coeruleus [47,61]. The main cholinergic afferents to the nucleus basalis arrive from the midbrain pedunculopontine (PPT), the lateral dorsal tegmental (LDT) nuclei, and the limbic cortex [62,63,64]. There is also GABAergic input in the form of symmetric synapses, which could represent local inhibitory neurons or projections from other brain regions [65,66,67]. It has been shown in multiple studies that the cholinergic neurons originating in NB and projecting to the cortex are regulated through GABAergic receptors [47,61,65,66,67]. There is also extensive bidirectional communication between the nucleus basalis and the prefrontal cortex, and in mice, the rostral–caudal distribution of cholinergic cells in the basal forebrain is associated with projection to the superficial–deep layers of the ventral medial prefrontal cortex, respectively [49].

Both the PPT and LDT nuclei receive afferent projections from a widespread number of common structures, most predominately the reticular formation in the brainstem, the midbrain central gray region, and the lateral hypothalamus–zona incerta region [68,69]. Retrograde tracing studies have shown inputs arriving in the LDT nucleus from the midbrain reticular formation, the periaqueductal gray, the medial preoptic nucleus, the anterior hypothalamic nucleus, the perifornical and lateral hypothalamic areas, the premammillary nucleus, paraventricular hypothalamic nucleus, zona incerta, and the lateral habenular nucleus [46,70]. The PPT nucleus receives afferent inputs from a wide range of regions, reviewed in more depth by Martinez-Gonzalez in 2011 [71], but most notably including afferent connections from the cortex [69,72,73], various locations in the basal ganglia [69,74,75,76,77,78,79,80,81], the locus coeruleus [82], and the dorsal raphe [83].

The MS receives noradrenergic inputs from the locus coeruleus [45,84] and serotonergic inputs from the raphe nuclei [85], as well as additional inputs from several other brainstem nuclei [85]. Afferent projections to the DB have not been well characterized, partly due to its poorly defined margins. Some studies have shown afferent connections arriving from the supermammillary nucleus [86] and reciprocal connections with the CA2 subfield of the hippocampus [59].

#### 2.1.3. Outputs

The nucleus basalis is an important source of ACh to the cerebral cortex [43,51,52,56,57], with efferent cholinergic projections that terminate on both pyramidal and GABAergic cells [55]. The nucleus basalis also supplies several thalamic nuclei with ACh, including the intralaminar nuclei, medial dorsal nucleus, and reticular nucleus [44,87]. Additional cells arise from the nucleus basalis that terminate throughout the amygdala, though most of these projections are GABAergic, with a minority being cholinergic [88,89]. 

The cholinergic nuclei in the midbrain, the PPT and LDT, have primary outputs that project to the nucleus accumbens, hypothalamus, raphe, and pontine and medullary reticular formations. They also project to the nucleus basalis, all thalamic nuclei, the amygdala, and the primary visual cortex [48,90]. The PPT, specifically, has long been thought to act as an interface between the basal ganglia and motor systems, though more recent work highlights the role of the varied neuronal subtypes and projections in updating behavioral states [91]. The LDT appears to play a significant role in activating the mesolimbic reward system [92]. 

### 2.2. Noradrenergic System

#### 2.2.1. Sources

Noradrenergic projection to the forebrain is exclusively provided by a single source, the locus coeruleus (LC), which is a small, bilateral nucleus located in the pons [93,94,95,96,97]. A complete review of the LC was provided by Poe et al. in 2020 [98], but a brief description is provided here. Traditional investigations of the LC presumed it to be a broadly acting, primarily homogenous source of NE with wide implications [93,94,99,100], but more recent research has shown that the LC is composed of many distinct modules with highly specific functional roles throughout the brain [98]. There are two major, complementary theories on how a diffusely projecting single source of norepinephrine can achieve such disparate functional results. The first is that the function of NE release relies on regional differences in postsynaptic receptor distribution and resulting differences in spatiotemporal NE reuptake [101,102,103]. The second is a corollary to the function of the noradrenergic system in the periphery, in which the sympathetic nervous system has discrete efferent limbs that are organ specific but capable of acting in a unified manner [104,105]. In this theory, the LC provides localized neuromodulation to well-defined target regions and spiking is synchronized in highly specific subsets of LC neurons. For a more complete review, see Totah et al., 2019 [106].

#### 2.2.2. Inputs

An important step in understanding the regional and modular functionality of the LC was achieved through an in-depth characterization of the afferent and efferent projections to and from the LC. The LC itself consists of a small, dense core, where cell bodies are found, and a peri-LC shell in which LC dendrites reside [107,108,109]. There are prominent afferent inputs to the LC core originating from the paragigantocellularis nucleus and the prepositus hypoglossi nuclei—both structures in the rostral medulla [110]. There are also additional inputs from the insular cortex, central nucleus of the amygdala, preoptic area, and the lateral and paraventricular hypothalamic areas [108,111,112]. Cerebellar Purkinje cells and neurons from deep cerebellar nuclei also provide synaptic inputs onto the core of the LC [108]. 

Although the projections of sensory afferents from the mesencephalic trigeminal sensory nucleus (Me5) [113,114] and the nucleus of tractus solitarius (NTS) [115] to the LC exert influences on cognitive functions [116], an important regulatory component on the core noradrenergic neurons in the LC include the peri-LC afferent innervations. Noradrenergic LC neurons possess long dendrites that pass through the surrounding small nuclei-like regions around the LC, which receive separate inputs from a variety of brain regions, including the prefrontal and infralimbic cortex, the amygdala, and the dorsal raphe nucleus [117]. There are additionally cholinergic, serotonergic, and adrenergic inputs to the peri-LC area, representing potential points of indirect regulation from other neuromodulatory systems [98,112]. The peri-LC zone also gives rise to several GABAergic inputs into the LC [118,119].

#### 2.2.3. Outputs

The efferent projections from the LC are widespread but nonuniform to the neocortex in both rodents [102] and primates [120,121]. Collateral axons from the LC are distributed in a coordinated fashion to target circuits with a specific function [98,108,122,123,124,125,126]. The efferent projections from the LC travel throughout the brain, providing NE input to the cortex, insula, hippocampus, thalamus, amygdala, and cerebellum. A full review of this system was provided by Schwarz and Luo in 2015 [21]. Though the projections are widespread, the selective activation of specifically patterned noradrenergic neurons is poorly understood and likely involves a complex interplay between inputs into the LC and interacting systems [21]. Nevertheless, it has been shown that genetically distinct groups of noradrenergic neurons project to regionally and functionally specific circuits [127]. Understanding the anatomically distinct efferent circuits underlying specific functional consequences is an ongoing area of research that will likely improve our understanding of the role of the LC in the context of localized function. 

As an important aspect of neuromodulation, the LC also directly projects to serotonergic, cholinergic, and dopaminergic nuclei, providing a centralized locus of control over, or feedback with, other neuromodulators [63,128].

### 2.3. Direct Communication between the Cholinergic and Noradrenergic Systems

Direct interactions between the cholinergic and noradrenergic systems are complex and likely highly dependent on regional context. Some example experiments have allowed an early understanding of some of these direct actions. Post-synaptic NE release inhibits approximately 90% of rat brainstem cholinergic neurons through the direct activation of inwardly rectifying K^+^ currents, most likely through the α_2_ adrenoreceptor [129]. In striatal cholinergic neurons, NE mediates depolarization through β_1_ adrenoreceptor activation [130]. It is also interesting to note that the LC has a unidirectional input into the basal forebrain, where a mix of α_1_ and β_1_ adrenoceptors are found on cholinergic neurons [131,132]. In the same region, however, GABAergic cells express α_2_, the activation of which suppresses neural activity. Overall, the net effect of LC modulation on the basal forebrain is the enhancement of ACh release in the cortex [21,133,134], though more sophisticated studies should be performed to determine the binding preference and net effect with varying levels of NE input [135].

Conversely, ACh acting on noradrenergic neurons is mediated primarily through α_3_ nicotinic receptors, leading to depolarization and NE release and the activation of the hypothalamic–pituitary–adrenal axis [136]. In the hippocampus, nicotinic receptors are also responsible for releasing NE from LC neuron terminals, likely through the NO/cGMP pathway [136,137]. Beyond evidence for local modulation of noradrenergic neurons by Ach [138,139], cholinergic receptors, presumed to be muscarinic, on LC neurons can act to centrally modulate LC firing. ACh induces increased firing rates, though the cholinergic source is not well understood.

These direct forms of interactive modulation demonstrate that even in the simplest scenarios in which cholinergic or noradrenergic activity occurs, the other system is being engaged. Whether or not this results in a synergistic or antagonistic effect likely depends on the extent to which each system is activated, the distribution of receptors in targeted regions, and the relative concentrations of each.

### 2.4. Indirect Communication of Cholinergic and Noradrenergic Systems

Cholinergic and noradrenergic projections to the prefrontal cortex are important for a variety of cognitive and executive functions. As such, neuromodulatory connections to distinct areas in the cortex such as the anterior cingulate cortex (ACC), medial prefrontal cortex (mPFC), and orbitofrontal cortex (OFC) control important aspects of an animal’s behavior [140,141,142,143]. An important study by Chandler et al. in 2014 showed that while both cholinergic and noradrenergic neurons projected to the cortex from their respective nuclei, their pattern of distribution varied across subregions in the prefrontal cortex [144]. Cholinergic neurons appeared to occur throughout all regions in a relatively equal distribution, while noradrenergic neurons projected to much more defined locations, which did not overlap with other monoaminergic projections [144]. 

Much of what is currently known about cholinergic and noradrenergic interaction has been researched in the context of various functional outcomes or neurological disease models. In a sheep model of chronic pain, concentrations of ACh and NE in the cerebral spinal fluid were measured and found to only be correlated to one another in those animals with pain [145]. In Alzheimer’s disease (AD), the close interplay between ACh and NE is being increasingly investigated under a theory that AD is a broad neuromodulatory disorder as opposed to a dysfunction of primarily the cholinergic system [34,146]. Locomotor activity, a process often disrupted in a wide array of neurological disorders, has also been shown to be mediated through cholinergic interaction with the noradrenergic system [147]. In attentional disorders, there is evidence indicating deficits in norepinephrine-mediated control of the cholinergic system in the parietal cortex [148].

An extremely important, though substantially under-researched, mode by which indirect interactions between these neuromodulatory systems occur is in their differential effect in glial cells such as astrocytes. Specifically, it has been shown that each neuromodulator has a specific effect on astrocyte potassium clearance, thereby regulating the extracellular potassium concentration and influencing local synaptic transmission [149].

There are also examples in the literature of the indirect engagement of the noradrenergic system that is likely mediated by α_7_ nicotinic receptors on GABAergic neurons and a resulting disinhibitory effect [150]. Supporting this is the direct measurement of increased ACh and NE in the rat cortex after administration of a nicotinic agonist [151]. It is important to note here that a simple increase and decrease in neurotransmitter level is not enough to discern the role of that change in a functional capacity. A more thorough understanding of the location, receptors, and other systems involved is needed to unravel functional consequences.

## 3. Role of Acetylcholine in the Brain

### 3.1. Major Cholinergic Receptor Subtypes and Function

The cholinergic system exerts its action by binding to two distinct receptor classes: nicotinic and muscarinic. Nicotinic acetylcholine receptors (nAChRs) are composed of five subunits arranged around a central core, forming a transmembrane channel that conducts Na^+^, K^+^, and Ca^2+^ when bound to ACh, leading to a local membrane depolarization [152]. The assembled receptor is constructed using two primary families of α and β subunits [152]. The standard neuronal configuration includes combinations of α2 through α6 and β2 through β4 proteins [153,154]. There are additional homomeric configurations composed of α7 through α9 subunits [155]. The central nervous system appears to be predominately composed of α4β2 or α7 nAChRs, of which the former has a higher affinity for ACh [156]. In the adult rat brain, there is little anatomical overlap between the heteromeric and homomeric nAChR types, indicating two distinct modes of ACh activity within distinct anatomical regions [157]. Overall, nAChRs are distributed widely but sparsely throughout the hippocampus and cortex at both pre- and postsynaptic locations [152] and expressed on interneurons, pyramidal cells, and stellate cells [158,159,160,161]. A series of previous experiments have shown that layers I, III, and V in the human cortex exhibit the highest binding of ACh. This was slightly different in the primary somatosensory cortex, where binding in layer III was highest, and in the primary motor cortex, where layers III and V were the highest [162]. The α7 receptors are expressed broadly across almost all glutaminergic and GABAergic neurons, though regional differences have been noted. The presence of a presynaptic nAChR almost universally results in an increased neurotransmitter release, across multiple types of neurotransmitters [152,163]. 

Muscarinic receptors (MAChR) are expressed throughout all layers of the cortex, though layers II and V exhibit the highest concentration [164]. There is a total of five known muscarinic receptor subtypes: M_1_ to M_5_. In general, M_1_ receptors are most abundant in the neocortex, hippocampus, and striatum [165]; M_2_ receptors are located throughout the entire brain [166]; M_3_ receptors only have a low level of expression throughout the brain [164]; M_4_ receptors are localized in the striatum [167]; and M_5_ receptors are also widely distributed across the brain [118]. These receptors have a seven-transmembrane region that is highly conserved in G-protein coupled receptors (GPCRs) and activates multiple intracellular signaling pathways, including phospholipase C (by M_1_, M_3_, M_5_), the inhibition of adenylyl cyclase (by M_2_ and M_4_), and the regulation of several ion channels [168,169]. MAChRs also activate mitogen-activated protein kinases (MAPKs), which regulates cell survival, differentiation, and synaptic plasticity [170,171,172]. 

### 3.2. Cholinergic Involvement in Learning and Decision Making

The cholinergic system plays an important role in higher cognitive functions, specifically in decision making and the learning process. Neuromodulation by acetylcholine is generally orchestrated through the differential activation of the nicotinic and muscarinic receptors. The nicotinic receptor, for example, has been shown to be easily desensitized and upregulated through the presence of nicotine, inducing long-term alterations in the decision-making process [173]. Nicotine has also been implicated in increasing impulsivity and disinhibition in decision-making [174]. Studies have also shown how nAChRs seem to be important in adapting appropriate choices to a specific outcome [175]. Specifically, the nAChR α7 receptor has been implicated in slowing learning rates in mice during knockout experiments [4], as well as cognitive improvement during enhanced activation [176]. 

Both nAChRs and MAChRs have been identified to contribute to risk and uncertainty [177], with several studies showing that a complex interaction between these two receptors contributes to cholinergic interneuron patterned activity [178,179]. This interaction seems to have a larger implication for network dynamics across different brain regions. Specifically, studies where MAChRs or nAChRs were activated or blocked demonstrated changes in neural synchrony across multiple EEG bands, invoking the emergence of theta-gamma coupling in the cortex and exhibiting a correlation to increased learning performance in item–context association behaviors [180,181]. 

Acetylcholine on a global level in the brain has been found to be relevant in almost every decision-making paradigm. Lesioning studies of the basal forebrain have shown disruptions in reversal learning in marmosets [182], as well as being implicated in memory storage [183,184]. The role of ACh in memory has been shown to act as a modulator of update speed and as a controller of meta learning [185]. Here, ACh modulates different neural systems throughout learning, regulating the appropriate amount of ACh in specific brain regions to appropriately and effectively learn and formulate memories [186]. 

### 3.3. Cholinergic Involvement in Attention

Cholinergic release primarily mediates attentional processing in the brain [17,187]. In a five-choice serial reaction time task (5-CSRTT), the nicotinic facilitation of attention was found to exist, and the magnitude of this facilitation was dependent on the level of attentional engagement [188]. In addition, by using the 5-CSRT, Robbins and colleagues demonstrated that nicotinic β2 subunits in the prelimbic cortex are crucial for mice to successfully detect a cue [189]. In an operant sustained attention task (SAT), both the detection of signals and the attentional performance were enhanced by the α4β2 nAChR agonist-evoked ACh increases in the mPFC [190]. In a knockout study, it was shown that task performance, which relies on highly attentive control, was impaired in a group of mice lacking the β2 subunit in the mPFC, in comparison with their wild-type littermates [191]. Similarly, the genetic deletion of this special subtype of nAChRs also results in compromised performance in an auditory discrimination paradigm [190], suggesting its critical role in selective auditory attention. 

In addition, there is evidence implying that the muscarinic system plays a role in directing attentional selection mechanisms [192,193,194]. Specifically, muscarinic receptors are believed to primarily modulate higher-level visual stimulus processing [194]. Attending to the receptive field of certain V1 neurons evokes an increase in these neurons’ firing rates, and scopolamine, a muscarinic antagonist, reduces this attentional modulation [195]. Interestingly, nicotinic antagonist does not exert a systematic effect [195]. Yet another aspect which can also be associated with attentional control is adaptive behavioral control [196]. Various genotypes of M_2_ modulate the high-level inhibitory control processes that require the processing of prior information and suppression of irrelevant information [196].

## 4. Role of Norepinephrine in the Brain

### 4.1. Major Noradrenergic Receptor Subtypes

The noradrenergic system exerts influence over brain function through three receptor classes: α_1_, α_2_, and β receptors. Each of these receptors has control over specific processes of neurotransmission and sympathetic nervous system regulation. α_1_ receptors are members of the adrenoreceptor family, a subset of G-protein coupled receptors [197]. They have been further classified into three distinct subtypes: α_1A_, α_1B_, and α_1D_. Each subreceptor has demonstrated unique quantitative differences in effect [197]. Several experiments have explored the different concentrations of these subtypes throughout the brain. Specifically, it has been shown that α_1B_ was more prominent in the thalamus, lateral amygdaloid nuclei, and cortical laminar areas, while α_1A_ was higher in the entorhinal cortex, amygdala, and general cerebral cortex areas [198]. Furthermore, transgenic mouse experiments have allowed for specific receptors to be knocked out, uncovering that both α_1A_ and α_1B_ have a similar expression throughout the central nervous system, just with different abundances [199]. Around 55% of the brain was shown to express α_1A_, 35% α_1B_, and less than 10% was found to express α_1D_ [200,201,202]. The function of α_1_ receptors is implicated in a variety of cognitive processes and synaptic efficacies. Beginning with synaptic involvement, α_1_ receptors have been shown to increase the firing frequency of pyramidal and somatosensory neurons of the visual cortex through the protein kinase C signaling (PKC) pathway [203,204]. They have also been implicated in the enhancement of glutamate and acetylcholine release as well as neuronal excitation via PKC pathways, calcium pathways, and excitatory synapses [205,206,207,208,209]. α_1_ has also been shown to affect non-neuronal function as well, with the modulation of synaptic transmission through astrocytes and glial cells [210,211,212]. With regards to cognitive functions, α1 receptors have been shown to be implicated in memory, motor and motivational behavior, memory retention, and storage, but most of these are associated with general norepinephrine release in the brain [213].

α_2_ receptors are also a type of G-protein coupled adrenoreceptor, classified into three subtypes: α_2A_, α_2B_, and α_2C_. Specifically, α_2_ receptors have been implicated in orchestrating the presynaptic inhibition of norepinephrine in the central and peripheral nervous system [214,215,216]. This inhibition is critical for regulation of normal involuntary processes including physiological functions of the heart, vision, and gastrointestinal systems. Using pharmacological agents such as prazosin or oxymetazoline, α_2A_ and α_2B_ receptors have been shown to have significant control over sympathetic outflow and blood pressure [216]. Several other studies have shown α_2A_ receptor agonists enhance both serotonin and norepinephrine release [216]. Interestingly, the abundance of α_2_ receptor subtypes is much more localized than α_1_. While literature here is limited, studies have shown that α_2B_ receptors are found almost exclusively in the thalamus, while α_2C_ receptors are found in the olfactory bulb, cerebral cortex, hippocampal formation, and dorsal root ganglia [217].

The final type of noradrenergic receptors, classified as β, are also a G-protein coupled receptor, divided into three subtypes: β_1_, β_2_, β_3_ [216]. There have been studies linking β receptors to synaptic plasticity, with norepinephrine acting on β receptors to dictate synaptic strength in hippocampal neurons, as well as NE released from the locus coeruleus enhancing LTD-related memory processing [218]. 

### 4.2. Noradrenergic Involvement in Learning and Decision Making

The noradrenergic system has been implicated in a variety of decision-making paradigms as well as throughout the learning process. Studies using optogenetics, pharmacological agents, and lesioning have brought to light the effect norepinephrine has on cognition and higher-order thought processes. One theory regarding the role of NE in decision making involves the idea of network reset, acting as an “internal interrupt” signal [219,220]. Here, it is explained that the phasic activation of locus coeruleus noradrenergic neurons causes an increase of NE throughout the cortex, invoking cognitive shifts and the potential reorganization of neural networks [221]. This shifted brain state is hypothesized to be better equipped for rapid behavioral adaptation and enhanced decision making [221]. Other theories point out how stimulus-induced firing patterns of the LC are closely attuned to behavioral performance, hypothesized from LC primate recordings in visual discrimination tasks [28]. Similar phasic activation in primates has shown how the LC can respond to specific task-related decisions, modulating NE release and adapting future task-relevant decisions [186], as well as showcasing coordinated activity patterns in cortical networks derived from ascending NE projections [222]. Studies invoking NE release through an agonist have shown enhancements in sensory stimulation, allowing more rapid synaptic plasticity and faster behavioral responses [223]. 

Several pharmacological experiments have investigated the specific role that α_2_ receptors play in the decision-making process. Studies using NE antagonists have shown α_2A_ receptor knockout leading to more risk-on behavior, with rats exhibiting greedier decisions [224]. α_2A_ agonists have been proven to enhance the efficiency of working memory and reduce impulsivity in primates [225]. This increased receptor uptake in the prefrontal cortex seems to be part of the shifted network brain state described earlier. The agonist guanfacine, another α_2A_ agent, was also shown to improve visual object discrimination performance during a reversal learning paradigm in primates [226]. 

### 4.3. Noradrenergic Involvement in Attention

The noradrenergic modulation of attention has been studied for several decades [227,228,229]. Studies have established the theory that the LC-NE system regulates the efficacy of information processing during neural coding of sensory signals [20,230,231]. During behavioral tasks, selective attention enhances neuronal responsiveness to sensory cues [232,233]. The firing rates of LC neurons are correlated with attentive behavior in an odd-ball task [227], in which either high or low tonic firing rates correspond to inattentive states and medium firing rates are associated with animals’ best performance. In a novel environment where more adaptive behaviors are required, changes in electrotonic coupling among LC neurons regulate goal-directed exploration and preserve attentional selectivity [28]. In addition, some studies have investigated the effects of NE agonists. It is shown that in a cued target detection task (CTD), the application of α_2_ receptor agonists clonidine or guanfacine significantly impaired alerting behavior, and the effect was dose-dependent [234], while the effect was blocked by the α_2_ antagonists idazoxan or yohimbine. 

Most recent studies also show an association between the NE system and impulsivity control [235,236,237]. It was observed from the superior frontal theta band activity that the NE system dynamically gains and loses relevance to regulate inhibitory control under different responding modes [237]. This work has led to the use of the NE-specific reuptake inhibitor atomoxetine as a treatment of pediatric attention-deficit/hyperactivity disorder (ADHD) [235]. Furthermore, it is demonstrated that ADHD patients have a higher positron emission tomography (PET)-measured NET availability in comparison to healthy individuals, suggesting that there are underlying genetic and epigenetic mechanisms.

## 5. Functional Interplay between the Cholinergic and Noradrenergic Systems

### 5.1. ACh and NE in Attention

It has been widely acknowledged that both cholinergic and noradrenergic systems show graded and transient increases in their response to increased attention to environmental cues [238,239,240]. Noradrenergic axon activity starts ~1 s prior to the peak of pupil dilation, while cholinergic axion activity lags ~0.5 s behind the peak, suggesting pupil-linked alertness, attention, and mental effort are controlled differentially by the two neuromodulatory transmitter systems [241]. Studies have shown that basal forebrain and brainstem cholinergic systems interact differently with the LC-NE system related to attention [42,242]. It was demonstrated that sustained attentional performance necessarily requires the integrity of BF cholinergic projections but not their noradrenergic afferents [243,244]. However, for thalamocortical information processing, ACh activation produces a noisy broadband signal detection mode, while NE activation sets to a noise-free high-frequency signal detection mode, which seems to be more optimized for selective attention than brainstem cholinergic activation [242]. 

Generally, cortical ACh–NE interaction plays a significant role in the modulation of attention [245,246,247]. Using fluorescent retrograde tracers in ACC, mPFC, and OFC, it was uncovered that subsets of LC neurons might be responsible for modulating individual prefrontal subregions independently, yet subsets of NB neurons might produce universal influence in prefrontal subregions [245], providing insight respecting prefrontal cortex’s role of allocating attentional reserves. In an attentional set shifting task, McGaughy and colleagues pointed out that the specific impairments in animals’ ability to shift attentional set were produced by noradrenergic instead of cholinergic deafferentation in the prefrontal cortex [247]. Indeed, the cortical cholinergic system is very likely to be involved in aspects of established attentional performance, while the NE system is more competent in detecting shifts in the predictive relationship between action and reinforcement [246]. 

Future studies are highly encouraged to explore the ACh–NE interaction in the contexts of their complementary roles regarding attention modulation. The systematic characterization and manipulation of the cholinergic and NE projections at biochemical, genetic, pharmacological, and physiological levels would largely facilitate our understanding of the interaction between the two systems and inform the development of potential therapeutics for certain neurodegenerative and psychiatric diseases.

### 5.2. ACh and NE in Learning and Decision Making

The interplay between the noradrenergic and cholinergic systems in the decision making and learning processes is highly complex. Each neuromodulatory system plays a pivotal role in creating these complex cognitive brain states, integrating sensory information with positive and negative feedback loops through multiple brain regions. For example, the visual system of a macaque primate has integration and circuitry across all neuromodulators, inhibiting and exciting different neurons, changing gene expression, and modulating synaptic circuitry [248]. Due to the complexity of decision making, the unpredictability of environments, and the uncertainty of risk-taking during exploration, the relationship between neuromodulators is not yet well understood [249]. Even with this challenge, there have been studies that have attempted to look at isolated NE and ACh interaction in specific decision-making tasks. In isolated behaviors, specific neuromodulators can be seen to regulate specific sub-tasks. In meta learning, NE can be seen to contribute to the randomness of action selection, while ACh seems to solely dictate the speed of specific memory updates [185]. Both neuromodulators have also been implicated in the information transmission during different behaviors [19,250]. Theoretical modeling has been used to further understand the role both NE and ACh play in uncertainty, behaving both synergistically and antagonistically, enabling complex learning in challenging adaptive environments [251] Understanding how both of these neuromodulators interact with one another in a variety of decision making and learning paradigms will be critical for developing translational treatments for neurological conditions that involve these neurotransmitters. 

## 6. Future Directions

The functional consequences of overlapping and interacting neuromodulatory systems are as numerous as they are behaviorally important. An understanding has been slowly emerging over at least two decades that a diversity of functionally distinct circuits and heterogeneously distributed receptor subpopulations between neuromodulatory systems gives rise to many of the most interesting aspects of neural processing and adaptive behavioral outcomes. However, the differential influence of each neuromodulator on a specific circuit of interest is highly complex and requires a very careful experimental framework in order to begin unraveling a more complete understanding of the influence of ACh and NE on specific behaviors. As has been demonstrated by the work covered in this review, overlapping, segmented receptive fields, non-uniform receptor distributions and the wide range of actions of ACh and NE, in a spatially and temporally dependent manner, make broad conclusions about each system difficult to generalize. It is critical for future work to use a modern arsenal of tools to dissect the role of these two systems in isolated circuits that have important behavioral relevance. 

Given the highly region-dependent action of these neuromodulatory systems, future work may begin with the identification and isolation of a specific target. An example of such an isolation would be to examine the role of the regions comprising the mPFC in decision making [175,252,253,254,255]. In order to understand the possible functional role of ACh and NE in modulating these regions, experiments should be devised to further characterize noradrenergic and cholinergic inputs to the mPFC. Initial experiments should also provide a basic characterization of the density and distribution of cholinergic and noradrenergic receptors in the target region using either traditional receptor expression profiling tools or the integration of new tools such as spatial single cell sequencing. Once the neuromodulatory input into the region of interest has been quantified and the range of receptors expressed summarized, an experimental framework can be established to systematically isolate one variable at a time.

In this framework, a functionally important region of the cortex can be isolated with respect to its neuromodulatory input, and the relevant molecular targets identified. At this point, the systematic exploration of the release of NE and ACh in a representative behavioral task will provide the foundation for understanding the differential presence of each system in the specified brain region during normal behavior. More sophisticated single-unit electrophysiology or calcium imaging experiments that explore the change in network dynamics in response to neuromodulator release can be paired with selective cholinergic and noradrenergic activation or silencing to observe the response not only of the animal and the neuronal connectivity but also the response of the other neurotransmitter system. Systematic knockouts or the implementation of inducible/repressible receptor expression in the specified region will also provide insights into how functional modules in the cortex are regulated by multiple neuromodulators.

The result of conducting many of these experiments in parallel will be an improved understanding of how broadly acting neuromodulator networks contribute to subdivided, and not necessarily unified, responses throughout the brain. Understanding the influence of ACh and NE on specific modules, and the afferent source of these neuromodulators, will provide improved targets for functional neural stimulation and greater degrees of freedom for higher bandwidth communication and dynamic network control through neural interfaces [256].

## 7. Conclusions

The cholinergic and noradrenergic modulation of functionally distinct regions of the brain has become one of the primary organizational principles behind understanding the contribution of each system to the diversity of neural computation in the central nervous system. Decades of work has shown that a diverse family of receptors, which stratify across different brain regions, and afferent and efferent projections that can be selectively activated, are critical in helping widespread neuromodulatory systems obtain substantial heterogeneity in the sophistication of their role in neural processes. The result of such a complicated interplay between two diffuse modulatory systems is a dynamic and highly context-dependent role for brain regions important in learning, memory, attention, and decision making. To further improve our understanding of these systems, it is essential to adopt approaches that are built on previous work to identify the interplay, at the receptor, circuit, and functional levels, between these systems in an isolated circuit that can be tied to a behaviorally functional outcome. The results will allow greater understanding and control over wide-ranging behaviors.

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
