# Peer review of "The Neuromodulatory Role of the Noradrenergic and Cholinergic Systems and Their Interplay in Cognitive Functions: A Focused Review"

_brainsci, 2022, doi:10.3390/brainsci12070890_

Round 1

Reviewer 1 Report

The authors provided a useful, interesting and comprehensive overview. Some formatting refinements are needed as listed below. Moreover, I have a some of suggestions about the paragraphs Noradrenergic system” > “Inputs” and “Noradrenergic involvement in attention”. No mention has been made about the input to the LC from the trigeminal system (more specifically the MesV), that is perhaps one of the most interesting and widely studied in the recent years (10.3389/fnana.2017.00130). It has also been demonstrated that the functioning of the LCs of both sides respectively, may have an effect on attentive tasks (10.1016/j.brainres.2020.147194 - 10.3389/fnsys.2021.749444). I believe that these evidences may add something in the explanation to the readers and may help focusing the attention on future researches.

Formatting refinements:

-        Lines 31-32: control the eventual formatting of superscript numbers

-        Control the formatting of citations

-        Control the formatting of paragraphs (it might be useful to provide an index at the beginning if in accordance to the journal policies)

Author Response

The authors provided a useful, interesting and comprehensive overview. Some formatting refinements are needed as listed below.

Thank you for your support.

Moreover, I have a some of suggestions about the paragraphs Noradrenergic system” > “Inputs” and “Noradrenergic involvement in attention”. No mention has been made about the input to the LC from the trigeminal system (more specifically the MesV), that is perhaps one of the most interesting and widely studied in the recent years (10.3389/fnana.2017.00130). It has also been demonstrated that the functioning of the LCs of both sides respectively, may have an effect on attentive tasks (10.1016/j.brainres.2020.147194 - 10.3389/fnsys.2021.749444). I believe that these evidences may add something in the explanation to the readers and may help focusing the attention on future researches.

Thank you for pointing us to these recent studies. We have integrated these important literatures into the revised manuscript.

Formatting refinements:

-        Lines 31-32: control the eventual formatting of superscript numbers

Thank you for pointing this out. The issues were caused by the conversion of our MS Word file to the publisher’s format. We will work with their production team to solve these issues. We have also asked the editor to forward our original MS Word file to the reviewers.

-        Control the formatting of citations

Again, the issues were caused by the conversion of our MS Word file to the publisher’s format. We will work with their production team to solve these issues.

-        Control the formatting of paragraphs (it might be useful to provide an index at the beginning if in accordance to the journal policies)

Thank you for your suggestion. We will talk to the publisher to see if it is possible to include an index in the main text in its final format.

Reviewer 2 Report

This is a clearly and well written review, and provides a useful overview over the neuromodulatory role of noradrenergic and cholinergic systems and their interplay. I do not see any major issues with this manuscript, in terms of quality or scope.

Author Response

This is a clearly and well written review, and provides a useful overview over the neuromodulatory role of noradrenergic and cholinergic systems and their interplay. I do not see any major issues with this manuscript, in terms of quality or scope.

Thank you for your support.

Reviewer 3 Report

On lines 31 and 32, the mathematics does not make sense. I am not sure if a zero is missing but it looks weird.

On lines 39 and 40, while the authors may be right about the total number of endogenous signaling molecules, based on classical definitions, there are only a handful of neurotransmitters. This is because of stringent criteria before a signaling molecule can be classified as a neurotransmitter. Anything else that is not a neurotransmitter is known as a signaling molecule.

On line 42, there is no such thing as “neurotransmitter receptors”. Just “receptors” would be enough.

On line 55, while the authors are not wrong that the scientific field does classify neurotransmitters as excitatory or inhibitory, it is worth noting that this classification is wrong. Neurotransmitters, themselves, are not excitatory or inhibitory. It is the receptors that determine if chloride ions or calcium ions go into the neuron or cell.

On line 74, this is not right. The ventral tegmental area produces dopamine but is not in either the basal forebrain or brainstem. The structures also has diverse and widespread projections to areas other than the cortex.

Author Response

On lines 31 and 32, the mathematics does not make sense. I am not sure if a zero is missing but it looks weird.

Thank you for pointing this out. The issue was due to misplacement of superscript numbers during the conversion of our MS Word file to the publisher’s format. We have asked the editor to forward our original MS Word file to the reviewers.

On lines 39 and 40, while the authors may be right about the total number of endogenous signaling molecules, based on classical definitions, there are only a handful of neurotransmitters. This is because of stringent criteria before a signaling molecule can be classified as a neurotransmitter. Anything else that is not a neurotransmitter is known as a signaling molecule.

Thank you for pointing this out. We have revised our statement and now it reads “Some of these substances, known as neurotransmitters, can act over varying physical distances, through their interaction with a much larger number of receptors”.

On line 42, there is no such thing as “neurotransmitter receptors”. Just “receptors” would be enough.

Thank you. We have replaced “neurotransmitter receptors” with “receptors” in the revised manuscript.

On line 55, while the authors are not wrong that the scientific field does classify neurotransmitters as excitatory or inhibitory, it is worth noting that this classification is wrong. Neurotransmitters, themselves, are not excitatory or inhibitory. It is the receptors that determine if chloride ions or calcium ions go into the neuron or cell.

Thank you for your comments. We have revised the manuscript to point out that neurotransmitters may have excitatory, inhibitory and neuromodulatory effects on neurons through the action of their receptors.

On line 74, this is not right. The ventral tegmental area produces dopamine but is not in either the basal forebrain or brainstem. The structures also has diverse and widespread projections to areas other than the cortex.

We apologize for the inaccurate statement in the original manuscript. We have revised our statement and now it reads “The groups of neurons responsible for delivery of these four neuromodulators are known as ascending neuromodulatory systems due to the fact each of these neurotransmitters originates in the brainstem, midbrain, or basal forebrain and projects to various brain structures.”